# Sector Piezoelectric Sensor Array Transmitter Beamforming MUSIC Algorithm Based Structure Damage Imaging Method

**DOI:** 10.3390/s20051265

**Published:** 2020-02-26

**Authors:** Tao Fu, Yi Wang, Lei Qiu, Xin Tian

**Affiliations:** 1Institute of Chemical Materials, China Academy of Engineering Physics, Mianyang 621900, China; futao@caep.cn; 2Research Center of Structural Health Monitoring and Prognosis, State Key Lab of Mechanics and Control of Mechanical Structures, Nanjing University of Aeronautics and Astronautics, Nanjing 210016, China; yiwang@nuaa.edu.cn

**Keywords:** structural health monitoring, elastic wave, sensor array, transmitter beamforming, multiple signal classification algorithm

## Abstract

Elastic-wave-based structural health monitoring technology has a broad application potential for its sensitivity and ability to achieve regional monitoring. For structures with large damping and specific shapes, the traditional damage monitoring method is limited by the sensor arrangement area and affected by low signal-to-noise ratios, so it is difficult to accurately locate the damage in a structure. To solve this problem, this paper proposed a damage monitoring method based on a sector piezoelectric sensor array for multiple signal classification algorithm. By arranging two sector piezoelectric sensor arrays that are suitable for a specific structure, the damage scattering array signal under the multi-excitation source was obtained and synthesized, the signal-to-noise ratios were improved, and the damage location accuracy was thus improved. The effectiveness of the method was verified by monitoring the damage in a circular bonded structure with a metal ring. Compared with the damage localization methods based on the traditional single excitation source multiple signal classification algorithm, path imaging and delay-sum imaging, this method can achieve better damage location and has a higher localization accuracy.

## 1. Introduction

High-reliability, high-performance structural safety services are of great importance to major or important mechanical equipment and infrastructure in fields such as aerospace, civil engineering, and rail transit. These services also challenge the traditional structural safety assessment and maintenance guarantee [1,2,3]. The application of structural health monitoring (SHM) technology, which has gradually become a popular research area, can effectively prevent major accidents, achieve the maintenance of conditions, and reduce maintenance costs [4,5,6,7]. Among the SHM methods, the elastic-wave-based SHM method is regarded as a promising damage monitoring method because it is sensitive to metal cracks and the impact damage of composite materials, and can be used to monitor structures in real time. It is considered to be a promising damage monitoring method [8,9,10,11,12,13,14,15]. Piezoelectric sensors are commonly used to excite and receive elastic waves in structures, and thus, damage monitoring in structures can be combined with the damage imaging method [16]. According to different sensor arrangement methods, damage imaging methods can be divided into sparse sensor array imaging and dense sensor array imaging.

In the research on imaging methods based on sparse sensor arrays, Michaels et al. [17] used a multisensing method to acquire structural response signals and image fusion technology to realize hole damage imaging in aluminum plates. Yuan et al. [18] combined the delay-sun imaging method with the probabilistic hybrid model and proposed a 4D damage imaging method. The damage imaging of a carbon fiber reinforced structure under the influence of temperature is realized. Chang et al. [1] proposed an energy-factor-based damage factor calculation method for each excitation-sensing path. The delamination damage of a reinforced composite plate and the crack damage of a metal aluminum plate were identified. Qing et al. [19] proposed a damage factor calculation method based on signal cross-correlation and introduced intelligent sandwich technology to monitor the corrosion of metal pipelines. According to the phenomenon that damage has the greatest influence on the direct wave packet, Su et al. [20] used image fusion technology to image all possible points in a structure and introduced the cumulative distribution function to achieve damage prediction. Yuan et al. [21,22] proposed a time reversal method based on the dual PZT arrangement, which could effectively eliminate the influence of the environment and boundary reflection and realize nonreference damage imaging. The team then studied a type of high spatial resolution imaging based on virtual time reversal, which could clearly display both the single damage and the dual adjacent damage in the plate. Shen et al. [23] proposed an enhanced Lamb wave virtual time reversal technique for damage detection with transducer transfer function compensation. The method can achieve more accurate and robust damage imaging results. Fan et al. [24] used a self-designed tomographic imaging system based on a piezoelectric sensor array to achieve an accurate reconstruction of corrosion damage in the plate structures. Shen et al. [25] used tomographic imaging technology to monitor the cracks in the holes of structures and compared the tomographic imaging performance under different sensor array configurations.

However, the sparse array layout will be affected by region constraints and structural complexity. As a result, the sparse sensor arrangement is no longer applicable, and the localization accuracy of the algorithm is reduced. With the deepening of research, elastic wave damage monitoring methods based on dense sensor arrays and related array signal processing methods have attracted more and more attention [26,27,28]. As a new array signal processing method, the multiple signal classification (MUSIC) algorithm combined with the elastic wave monitoring method has been introduced for structural health monitoring due to its directional scanning capability and the potential of multisource monitoring [29,30]. Early MUSIC algorithms were mostly used for the angular localization of impact sources [31,32,33,34,35]. In recent years, Zhong and Yuan [36] proposed a near-field MUSIC algorithm based on a linear sensor array, which solved the near-field dead zone problem of the far-field MUSIC algorithm and verified the damage location on the composite plate structure.

For some high-damping materials or high attenuation structures with special structural forms, the elastic wave attenuates rapidly in the process of propagation, which means the signal-to-noise ratios (SNRs) of the damage scattering signals will be low, and thus, the damage location accuracy will be affected. For circular pipeline structures, Golato et al. [37] proposed a sparse array pipeline surface damage monitoring method based on two circular sensor rings. In this method, the linear signal model is reconstructed by grouping, and the damage location in a circular pipeline structure is realized. Chattopadhyay et al. [38] introduced elastic waves in their study and used space–time analysis without a reference method to achieve the damage localization of an X-COR sandwich high attenuation composite structure. Qu et al. [39] proposed a new damage detection technique for use in highly damped composite structures using the time reversal method with different excitation amplitudes. Qing et al. [40] proposed a damage monitoring method for torsional elastic waves, which realized the monitoring of high attenuation honeycomb structures. Bao et al. [41,42] proposed the transmitter beamforming MUSIC algorithm based on a bilinear array and achieved low SNRs corrosion damage monitoring in a reinforced aviation structure. The array error was also corrected, and thus, the localization accuracy of the algorithm was improved. The algorithm was verified on an anisotropic high attenuation composite reinforced structure.

In summary, the damage imaging method based on a sparse sensor array is susceptible to structural limitations, while the single excitation source MUSIC damage location method based on a dense sensor array will lead to localization errors due to the problem of low SNRs. In this paper, a transmitter beamforming MUSIC damage imaging method based on a sector sensor array is proposed to perform damage monitoring of a special circular bonded structure. In this method, according to the specific form of the circular bonded structure, the sector-shaped double sensor array is arranged, and the damage scattering array signal is superimposed at the search point according to the time delay rule to improve the SNRs. Through the eigenvalue decomposition of the covariance matrix of the superimposed sensor array signal, direction scanning is carried out on the structure, and the spatial spectrum of the monitoring area is constructed to locate the damage.

The paper consists of the following parts. Section 2 introduces the basic principle and damage monitoring process of the sector piezoelectric sensor array transmitter beamforming MUSIC (SPATB-MUSIC) algorithm. Section 3 studies the propagation properties of elastic waves in a circular bonded structure with a metal ring. In Section 4, the damage location experiment was performed on a circular bonded structure, and the localization details and results are given. Section 5 presents the conclusion.

## 2. SPATB-MUSIC Algorithm

In this section, based on the transmitter beamforming and MUSIC algorithm, a SPATB -MUSIC algorithm is introduced in detail. The method uses a dual array consisting of two sector sensor arrays. The method then produces an elastic wave by driving one of the sensor array rows and receives elastic waves with the other sensor array row. Then, in the search for the monitoring area, the focus signal is obtained by making the elastic wave of each element in the sensor array reach the damage at the same time. The focus array signal is used as the input of the MUSIC algorithm to obtain the noise subspace and the steering vector space, the spatial spectrum is used to describe the orthogonality between them, and the damage location is realized by jointly searching the entire monitoring region.

### 2.1. Elastic Wave Signal Propagation Model under Sector Piezoelectric Array

In the sound source estimation method, it can be divided into far field and near field, according to the distance of the sound source to the sensor array. In the far field model, the wavefront can be approximated as the linear wavefront, and the location of damage can be determined only by searching the angle. However, for the structure monitored in this paper, the distance between the damage and the sensor array is short. If the approximation is made in the far field, a large positioning error will occur. Therefore, the near field model is introduced. Figure 1 shows the near field elastic wave propagation model of a single sector piezoelectric array. There are a total of 2*M* + 1 elements, which are located on the circumference of the radius *R*, and the distance between the adjacent elements is the same. The coordinate system is established by taking the central array element as the coordinate origin and the reference array element. The coordinates of the *i*th array element are (*R_i_, w_i_*), where *R_i_* is the distance between the *i*th array element and the reference array element, and *w_i_* is the clip with the x-axis angle. In the near-field case, the wavefront cannot be approximated as the linear wavefront but as the circular wavefront, and the distance between the signal source and the array elements of the sensor array is different, while the angles of the array elements with respect to the *x*-axis are not consistent.

If the center frequency of the signal is *ω*_0_, the response signal *x*_0_(*t*) of the reference element PZT_0_ can be expressed as
(1)$$x_{0}(t)\;=\;u(t)e^{j(\omega_{0}t-kr)}$$
where *u*(*t*) is the damage scattering signal; *r* is the distance from the damage to the reference array element; *k* is the wavenumber (*k* = *ω*_0_/*c*); and c is the elastic wave propagation velocity. Then, the response signal *x*_i_(*t*) of the array element PZT*_i_* in the array can be represented as
(2)$$x_{i}(t)=x_{0}(t)e^{j\omega_{0}\tau_{i}}+n_{i}(t),i=-M,\cdots ,M$$
where *n_i_*(*t*) is the background noise, and *τ_i_* is the time difference caused by different arrival times of the damage scattered signal due to the different distances between PZT*_i_* and PZT_0_ from the damage. The method of calculation is as follows:(3)$$\tau_{i}=\frac{\Delta r_{i}}{c}=\frac{r-r_{i}}{c}$$

The path propagating from the signal source to PZT_0_ and PZT*_i_* and the connection with PZT_0_ and PZT*_i_* can form a triangle. According to the cosine theorem, the distance *r_i_* from the signal source to each element of the piezoelectric sensor array can be expressed as
(4)$$r_{i}=\sqrt{r^{2}+R_{i}{}^{2}-2rR_{i}\cos(\theta -w_{i})},i=-M,-M+1,\cdots ,M$$
and *r_i_* is not parallel to *r*. In this equation, *θ* is the angle between the damage source and the reference element and the *x*-axis, and in Equation (4) instead of Equation (3), the time delay can be rewritten as
(5)$$\tau_{i}=\frac{r-\sqrt{r^{2}+R_{i}{}^{2}-2rR_{i}\cos(\theta -w_{i})}}{c},i=-M,-M+1,\cdots ,M$$
From this equation, it can be found that the time delay is related not only to the sound source angle, but also to the distance. The guidance vector of the signal source in the near field case is defined as
(6)$$a_{i}\left(r,\theta \right)=e^{-j\omega_{0}\tau_{i}}$$
Writing the array response signal in the form of a vector,
(7)$$\begin{array}{ll} & X\left(t\right)={\left[\begin{array}{cccc} x_{-M}\left(t\right) & x_{-M+1}\left(t\right) & \cdots & x_{M}\left(t\right) \end{array}\right]}^{T}\\ & A\left(r,\theta \right)={\left[\begin{array}{cccc} a_{-M}\left(r,\theta \right) & a_{-M+1}\left(r,\theta \right) & \cdots & a_{M}\left(r,\theta \right) \end{array}\right]}^{T}\\ X\left(t\right)=A\left(r,\theta \right)s_{0}\left(t\right)+N\left(t\right),\; & N\left(t\right)={\left[\begin{array}{cccc} n_{-M}\left(t\right) & n_{-M+1}\left(t\right) & \cdots & n_{M}\left(t\right) \end{array}\right]}^{T} \end{array}$$

### 2.2. Transmitter Beamforming under Dual Sector Piezoelectric Array

In practical applications, for the bonding structure monitored in this paper, the thickness and damping are large, and the SNRs of the damage scattering signal is small, which leads to the error of the final localization result. To solve this problem, a double sector piezoelectric array was used, and a transmitter beamforming MUSIC algorithm based on the sector piezoelectric array was proposed. By arranging double sector array sensors, the distance damage delay of the excitation array elements can be calculated. The damage scattering signal is superimposed to enhance the signal and effectively improve the SNRs of the damage scattering signal.

As shown in Figure 2, the elastic wave propagation models for arranging the double sector array sensors are named as array A and array S, respectively, where array A is the excitation source array, and array S is the sensor array. The number of elements of each sector array is 2*M* + 1, each element is named from left to right, the excitation source array is A_−*M*_ to A*_M_*, and the sensor array is S*_−M_* to S*_M_*. The coordinates of element *p* in the excitation array are $\left(R_{p}^{A},w_{p}^{A}\right)$, and $r_{p}^{A}$ is the distance from the *p*th array element of the excitation source array to the damage.

Therefore, according to Equation (8), the distance from each element of the excitation source array to the damage location and the time delay of each element signal arriving at the damage relative to the reference element signal can be calculated.
(8)$$\begin{array}{l} r_{p}^{A}=\sqrt{{\left(R_{p}^{A}\cos w_{p}^{A}-r\cos\theta \right)}^{2}+{\left(R_{p}^{A}\sin w_{p}^{A}-r\sin\theta \right)}^{2}}\\ t_{p}=\frac{\Delta r_{p}^{A}}{c}=\frac{r_{p}^{A}-r_{0}^{A}}{c},p=-M,\cdots ,M \end{array}$$

The focused damage scattering signal can be represented by Equation (9). *X_p_*(*t*) is the damage scatter signal when the *p*th array element is excited in the array. At this time, *X*(*t*) is an enhanced damage scattering array signal.
(9)$$X\left(t\right)=X_{-M}\left(t\right)e^{-j\omega_{0}t_{-M}}+\cdots +X_{p}\left(t\right)e^{-j\omega_{0}t_{p}}+\cdots +X_{M}\left(t\right)e^{-j\omega_{0}t_{M}}$$

### 2.3. MUSIC Algorithm

The eigenvalue decomposition is performed by using the enhanced array signal as an input of the MUSIC algorithm as shown in Equation (10):(10)$$R=E\left\{XX^{H}\right\}=AE\left\{SS^{H}\right\}A^{H}+E\left\{NN^{H}\right\}=AR_{S}A^{H}+R_{N}^{}$$
***R****_S_* and ***R****_N_* are the signal covariance matrix and the noise covariance matrix, respectively. Assuming that the noise power is *σ*^2^, Equation (10) is transformed into
(11)$$R=AR_{S}A^{H}+R_{N}^{}=AR_{S}A^{H}+\sigma^{2}I$$

The eigen decomposition of the array signal covariance matrix is as follows:(12)$$R=U\Sigma U^{H}$$
where ***U*** is an eigenvector matrix, and ***Σ*** is a diagonal matrix composed of eigenvalues, as shown in Equation (13).
(13)$$\Sigma =\left[\begin{array}{cccc} \lambda_{1} & & & \\ & \lambda_{2} & & \\ & & \ddots & \\ & & & \lambda_{2M+1} \end{array}\right],\lambda_{1}\geq \lambda_{2}\geq \cdots \geq \lambda_{K}\geq \lambda_{K+1}=\cdots =\lambda_{2M+1}=\sigma^{2}$$
The diagonal matrix composed of large eigenvalues is defined as ***Σ_S_*,** and the diagonal matrix composed of small eigenvalues is defined as ***Σ_N_***.
(14)$$\Sigma_{S}=\left[\begin{array}{cccc} \lambda_{1} & & & \\ & \lambda_{2} & & \\ & & \ddots & \\ & & & \lambda_{K} \end{array}\right]\Sigma_{N}=\left[\begin{array}{cccc} \lambda_{K+1} & & & \\ & \lambda_{K+2} & & \\ & & \ddots & \\ & & & \lambda_{2M+1} \end{array}\right]$$

The eigenvector corresponding to ***Σ_S_*** is then tensioned into a signal subspace ***U****_S_* = [*e*_1_
*e*_2_
^…^
*e_k_*], the eigenvector corresponding to ***Σ_N_*** is tensioned into a noise subspace ***U****_N_* = [*e_k+_*_1_
*e_k+_*_2_
^…^
*e*_2*M*+1_], and Equation (12) can be further written as Equation (15).
(15)$$R=U_{S}\Sigma_{S}U_{S}^{H}+U_{N}\Sigma_{N}U_{N}^{H}$$
As the signal and noise are independent of each other, the noise subspace and the signal subspace are orthogonal, and the guidance vector of the signal subspace is also orthogonal to the noise subspace.
(16)$$A^{H}\left(r,\theta \right)U_{N}=0$$

Due to the existence of the background noise, ***A***(*r*, *θ*) and ***U****_N_* are not completely orthogonal. Therefore, Equation (17) is used to realize the minimum optimal search location.
(17)$$\left(\hat{r},\hat{\theta }\right)=\mathrm{argmina}A^{H}\left(r,\theta \right)U_{N}U_{N}^{H}A\left(r,\theta \right)$$
Therefore, the spectral estimation equation of the MUSIC algorithm is defined as
(18)$$P_{\mathrm{MUSIC}}\left(r,\theta \right)=\frac{1}{A^{H}\left(r,\theta \right)U_{N}U_{N}^{H}A\left(r,\theta \right)}$$

The entire monitoring area is searched to obtain the spatial spectrum on the monitoring area. When the searched location is the damage location, a spectral peak will appear in the spatial spectrum because the strongest array scattering signal is obtained. The damage location can be obtained by Equation (19).
(19)$$\left(r_{d},\theta_{d}\right)=\mathrm{argmax}P_{\mathrm{MUSIC}}\left(r,\theta \right)$$

In summary, the basic procedure of the damage monitoring method of the transmitter beamforming MUSIC algorithm based on the sector piezoelectric array can be divided into the following four parts, as shown in Figure 3:(1)The signal of the structural damage scattering array is obtained.(2)By using the transmitter beamforming equation, the damage scattering array signal is obtained.(3)The synthesized damage scattering array signal is decomposed by eigenvalue decomposition, and the orthogonal signal subspace and noise subspace are obtained.(4)The spatial spectrum of the corresponding monitoring region is obtained by spectral peak searching, and the corresponding coordinates of the spectral peak are the results of the damage location.

## 3. Propagation Characteristics of Elastic Waves in a Circular Bonded Structure

In this section, the circular bonded structure is introduced in detail. The elastic wave propagation characteristics experiment in this structure is described in detail. The amplitudes and wave velocities of the signals in a typical sensing channel are counted.

### 3.1. Circular Bonded Structure and Experimental Setup

The experimental setup includes an integrated elastic wave signal scanning system and a circular bonded structure with a smart piezoelectric sensor array layer attached, as shown in Figure 4. The integrated scanning system is used to generate an excitation signal and to obtain a response signal from the PZT sensor output [43]. In the experiment, a sinusoidal modulation five-peak signal was used as the excitation signal. The amplitude was ±70 V, the central frequency was set to 50 kHz, and the sampling rate was set to 10 MSPS. A schematic view of the dimensions of the structure, which is formed by bonding plexiglass and steel with a two-component epoxy adhesive, is shown in Figure 5. The plexiglass part was cylindrical and had a thickness of 35 mm and a radius of 45 mm. There were cylindrical grooves with a radius of 15 mm, and a hemispherical groove with a radius of 10 mm in the center. The inner diameter and outer diameter of the steel ring were 10 mm and 15 mm, respectively, and the thickness was 2 mm.

A group of piezoelectric sensors numbered from 1 to 18 were attached to the surface of the test piece. The diameter of the sensor was 8 mm, and the thickness was 0.48 mm. The center of the sensor is located on a circumference with a radius of 40 mm. During the monitoring process, sensors numbered from 1 to 7 and from 12 to 18 are selected to form a double sector array, named array A and array S, respectively. Each array contained seven piezoelectric sheets, and array A was used to excite the elastic waves. Array S was used as the sensing receiving elastic wave signal. The distance between adjacent sensors in the array was 9 mm, and a total of 49 sensing channel signals were obtained.

### 3.2. Propagation Characteristics of Elastic Waves in Circular Bonded Structures

This section investigates the propagation properties of elastic waves in a circular bonded structure. To obtain the propagation characteristics of the elastic wave in the structure, the elastic wave signals in the form of three channels were selected for analysis, as shown in Figure 6. The signals in the PZT A_1_-PZT S_18_ channel did not pass through the bonding and variable thickness region, while the signals in the PZT A_1_-PZT S_15_ channel passed through the bonding area, but did not pass through the variable thickness region. The signals in the PZT A_1_-PZT S_12_ channel passed through the bonded and variable thickness regions. Figure 7 shows the waveform in the form of three channels. The first wave packet of the signal was not analyzed due to the crosstalk caused by the line, followed by the signal direct wave. When the path length of the PZT A_1_-PZT S_18_ channel was only 62.4 mm, the direct band was more obvious, and the crosstalk was lightly aliased. When the path length of the PZT A_1_-PZT S_15_ channel was 75.5 mm, the signal direct wave was clear and no aliasing occurred. As the path length of the PZT A_1_-PZT S_12_ channel was 80.0 mm, the boundary forms were very complex. When the elastic wave propagates in the structure, it encounters the structure boundary, which results in the reflection of the signal and generates the boundary reflection wave. When the time difference between the direct wave and the boundary reflection wave is small, the waveform will be aliased, as shown in Figure 7c. However, the signals used in this paper were damage scattering signals; most crosstalk and boundary reflections were suppressed; and aliasing had little effect on the calculation of wave velocity. Therefore, the effect of aliasing on the final imaging results was small.

To further study the propagation law of elastic wave signals in the structure, the propagation velocity of the signal direct wave was calculated, and the Shannon plural wavelet transform method was used to obtain the excitation signal and response signal envelope [9]. As shown in Figure 8, the envelope of the PZT A_1_-PZT S_18_ channel signal was extracted, the flight time of the signal was obtained by the peak corresponding to time subtraction, and the signal propagation speed was calculated in combination with the direct path length of the signal.

The calculation results of the velocity are given in Table 1, and the amplitude of the direct wave crest of the signal was statistically analyzed. The amplitude of the signals in the PZT A_1_-PZT S_15_ channel, which passed through the bonded area without passing through the region of variable thickness, was 0.814 V. The amplitudes of the other two channels were only approximately half of their amplitudes at 0.486 V and 0.383 V, and the wave velocity was 699 m/s. The wave velocities of the other two channels were 722 m/s and 755 m/s, and the velocity difference was small. It can be seen that when the elastic wave signal propagates in this kind of bonding structure, the amplitude is small, the difference of the signal propagation speed is small, and the average velocity of signal propagation can be used for damage location in the process of monitoring. According to the above velocity calculation method, the average velocity of the 49 channels was 736 m/s, which was used for damage location in the next section.

## 4. Damage Location in Circular Bonded Structures

In this section, the SPATB-MUSIC algorithm is used to evaluate the damage in the circular bonded structure, and this method is compared with the traditional single excitation source MUSIC algorithm, the path imaging method, and the delay-sum imaging method.

### 4.1. Damage in the Structure and Monitoring

As shown in Figure 9, the part indicated by the red line is the location of the damage, and four damages were constructed in the structure by peeling the metal bonded to the plexiglass. As shown in Figure 10a, in undamaged conditions, the elastic wave array signal in the structure is obtained as a reference signal, and then, as shown in Figure 10b–e, the damage is artificially constructed in the structure in turn. The elastic wave array signal is obtained after the damage is generated at each place of the structure, and the signal in the former damage state is used as the reference signal in the latter damage state. The signal excitation acquisition system and parameter setting were the same as those in Section 3.1. The elastic waves were excited by the sensors in array A, and the sensors in array S were used to obtain the response signals under the 49 sensing channels.

Table 2 shows the damage information. A total of four damages were constructed, which were numbered D1, D2, D3, and D4. To verify the monitoring ability of the proposed method for different sizes of damage, dimensions of 10 × 5 mm^2^ and 5 × 5 mm^2^ were set. The dimensions of D1, D3, and D4 were 10 × 5 mm^2^, and the dimensions of D2 were only 5 × 5 mm^2^. The specific location of the damage is shown in the table.

### 4.2. Effect of Damage on the Signal

To study the influence of damage on the signal in the structure, Figure 11 shows the reference signals and damage signals for the PZT A_4_-PZT S_18_ and PZT A_4_-PZT S_14_ channels with damage sizes of 10 × 5 mm^2^ and 5 × 5 mm^2^, respectively. As the damage is located on the direct path of the signal, the signal given here is the direct part of the signal. It can be seen that when there was damage with a size of 10 × 5 mm^2^ in the structure, the damage signal changed with respect to the reference signal, and the amplitude increased. Although the damage scattering signal had clear wave packets, the SNRs were low. The amplitude only changed by approximately 0.070 V. For damage with a size of 5 × 5 mm^2^, the signal change was smaller, the damage scattering signals were aliased, and the wave packet was not obvious.

### 4.3. Transmitter Beamforming of Damage Scattering Array Signal

To study the beamforming effect, a comparison chart of the damage scattering array signals for damage sizes of 10 × 5 mm^2^ and 5 × 5 mm^2^ before and after enhancement is presented. As shown in Figure 12a, when the damage size was 10 × 5 mm^2^, PZT A_4_ was used as the excitation sensor, and array S was used to receive the array signals. Although there were obvious damage scattering segments and the wavefront was formed, the amplitude of the signal was less than that of the signal after the scatter segment. The damage scattering array signal shown in Figure 12b is the result of beamforming. It can be seen that the damage scattering segment was enhanced, and the PZT S_12_ receiving signal was removed. The remaining sensor receiving array signals had the largest amplitude of the damage scattering segment. As shown in Figure 13a, for a damage size of 5 × 5 mm^2^, the PZT A_4_ was used as the excitation sensor, and array S was used as the received damage scattering array signal. It can be seen that the amplitude of the damage scattering section was small at this time, and no obvious wavefront could be observed. The signal wave packets were mixed. Conversely, the damage scattering array signal obtained by transmitter beamforming enhanced the damage scattering section and formed a more obvious wavefront. However, time of flight (ToF) changed after beamforming, for which there are two reasons. (1) In the beamforming algorithm, the average speed of signal propagation in all channels is used to calculate the ToF when the signal reaches the damage. However, there are differences in the signal propagation speed in each channel, resulting in changes in ToF after beamforming. (2) Before beamforming, the damage sensitive segment of the damage scattering signal has a small amplitude and is aliased with the boundary reflections, while the array signal after beamforming is synthesized by seven groups of array signals, so that the damage sensitive segment is enhanced and becomes obvious, resulting in the visual changes of ToF before and after beamforming.

### 4.4. Damage Localization Result

Using PZT A_4_ as the excitation source, the damage scattering array signal obtained by the difference between the reference signal received by array S and the damage signal was used as the input of the MUSIC algorithm. The angle search range was from 0° to 180°, and the distance was from 0 cm to 8 cm. The lengths were set to 1° and 0.1 cm, respectively, and the results of the four damage locations are shown in Figure 14. The yellow circle is the actual damage location, and the maximum pixel value in the cloud image is the actual localization result. The localization errors were (3°, 0.8 cm), (1°, 1.0 cm), (11°, 0.1 cm), and (6°, 0.1 cm). It can be seen that there were errors in the location of the four damages. The angle error was up to 11°, the distance error was up to 1.0 cm, and the background noise in the 5 × 5 mm^2^ damage location cloud map was relatively large.

Figure 15 shows the four damage localization cloud maps using the SPATB-MUSIC algorithm. Table 3 shows the localization results and errors of the four damages. In this condition, sensor array A was used as the excitation source, and array S was taken as the reception. The four localization errors were (0°, 0.3 cm), (1°, 0.5 cm), (1°, 0.2 cm), and (0°, 0.2 cm). It can be seen that the localization accuracy for the damage was higher using the method, and the localization accuracy was greatly improved relative to the single excitation source. The angle error was only 1° at the maximum, the distance error was only 0.5 cm, and the background noise of the damage localization cloud image of 5 × 5 mm^2^ was effectively suppressed. 

The results show that this method could also accurately locate the damage of the structure in the case of a low SNRs.

The localization results of two of the more commonly used damage localization methods for the damage size of 5 × 5 mm^2^ are also given. One is the path imaging method combined with the damage factor, which assigns the probability of the points in the ellipse monitoring area formed by each path and then superimposes the attenuation factor extraction results of each sensing channel. The damage location imaging result is shown in Figure 16a. The localization result was (92°, 3.7 cm). This was 1.5 cm away from the actual location, and the error was 18.8% compared with the monitoring area diameter of 8 cm. The calculation method is as shown in Equation (20),
(20)$$Error_{\sigma }=\frac{Error_{d}}{R}$$
where *Error_d_* is the distance from the damage localization result and the actual damage location; *R* is the diameter of the monitoring area; and *Error_σ_* is the relative error. In the other method, the delay is calculated by using the propagation velocity of the elastic wave in the structure, and the wave packet amplitude of the damage scattering signal was stacked according to the time delay rule to obtain a pixel map reflecting the location of the damage. When the corresponding point of the actual damage location is searched, the pixel value increases abnormally, and thus, the damage location is realized. The localization result is shown in Figure 16b. The localization result was (90°, 6.3 cm) and the relative error was 13.8%. For comparison, the localization result of the STAPB-MUSIC method was also presented as shown in Figure 16c. It can clearly be seen that in the imaging diagram of the proposed method in this study, the pixel focus location was closer to the actual damage location. Table 4 shows the damage localization errors between the methods proposed in this paper and the above two imaging methods. It can be seen that the SPATB-MUSIC algorithm has a higher damage location accuracy than the two traditional imaging methods, with a relative error of only 6.3%.

## 5. Conclusions

The small signal variation due to damage and structural limitations causes difficulties in the traditional structural damage imaging method. In this paper, a structural damage monitoring method based on the SPATB-MUSIC algorithm was proposed for circular structures. In this method, a double sector piezoelectric sensor array is arranged on the surface of the structure. The elastic wave array signal under a multi-excitation source is obtained. The damage scattering array signal is obtained by subtracting the reference signal from the damage signal, and the transmitter beamforming method is then used to enhance the damage scattering signal and improve the signal SNRs. The enhanced damage scattering array signal is used as the input of the MUISC algorithm to realize the accurate location of the damage. The effectiveness of the method was verified by a damage monitoring experiment on a large circular damping structure formed by bonding plexiglass and metal materials. The experimental results showed that the proposed method has a higher localization accuracy than the traditional single transmitter source MUSIC algorithm, the path imaging method, and the delay-sum imaging method. The angular localization error was less than 1°, and the distance error was less than 0.5 cm. Although the STAPB-MUSIC method proposed in this paper uses the direct wave to locate the damage of the structure with high accuracy, there is still a lack of further research on the propagation characteristics of the signals and wave modes in the structure. In the future, we will carry out more detailed research on this aspect.

## Figures and Tables

**Figure 1 sensors-20-01265-f001:**
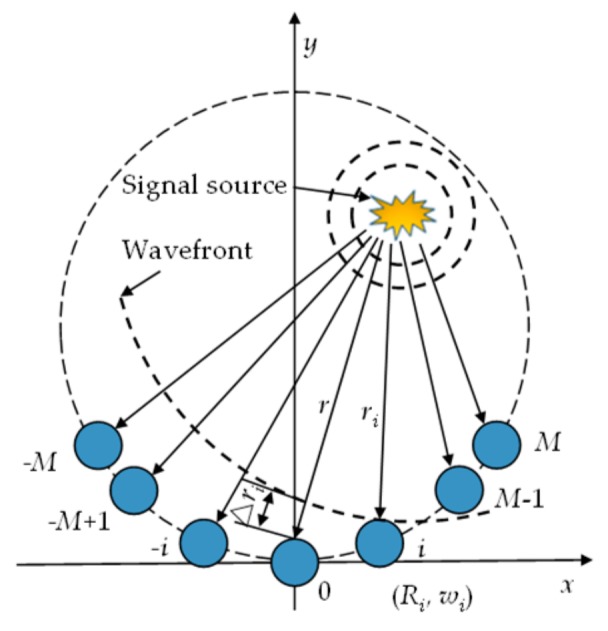
Elastic wave propagation model in the near field.

**Figure 2 sensors-20-01265-f002:**
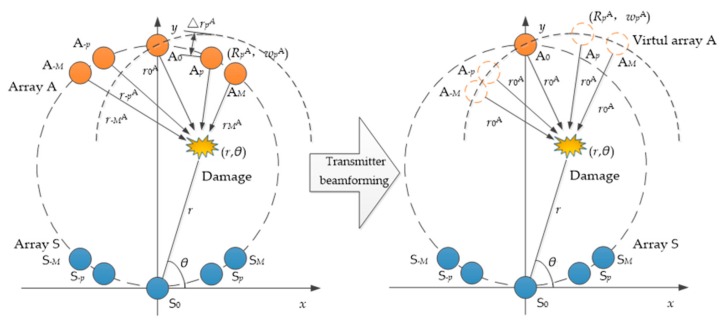
Transmitter beamforming based on the sector piezoelectric array.

**Figure 3 sensors-20-01265-f003:**
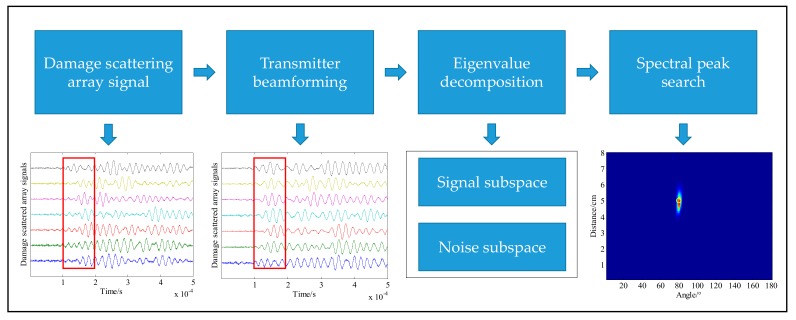
Basic flow of the SPATB-MUSIC damage monitoring method.

**Figure 4 sensors-20-01265-f004:**
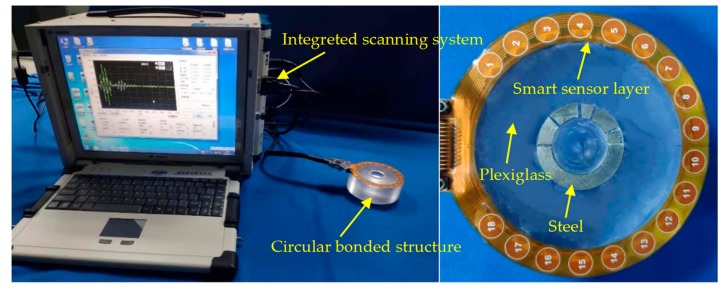
Experimental system and the circular bonded structure with a smart sensor layer.

**Figure 5 sensors-20-01265-f005:**
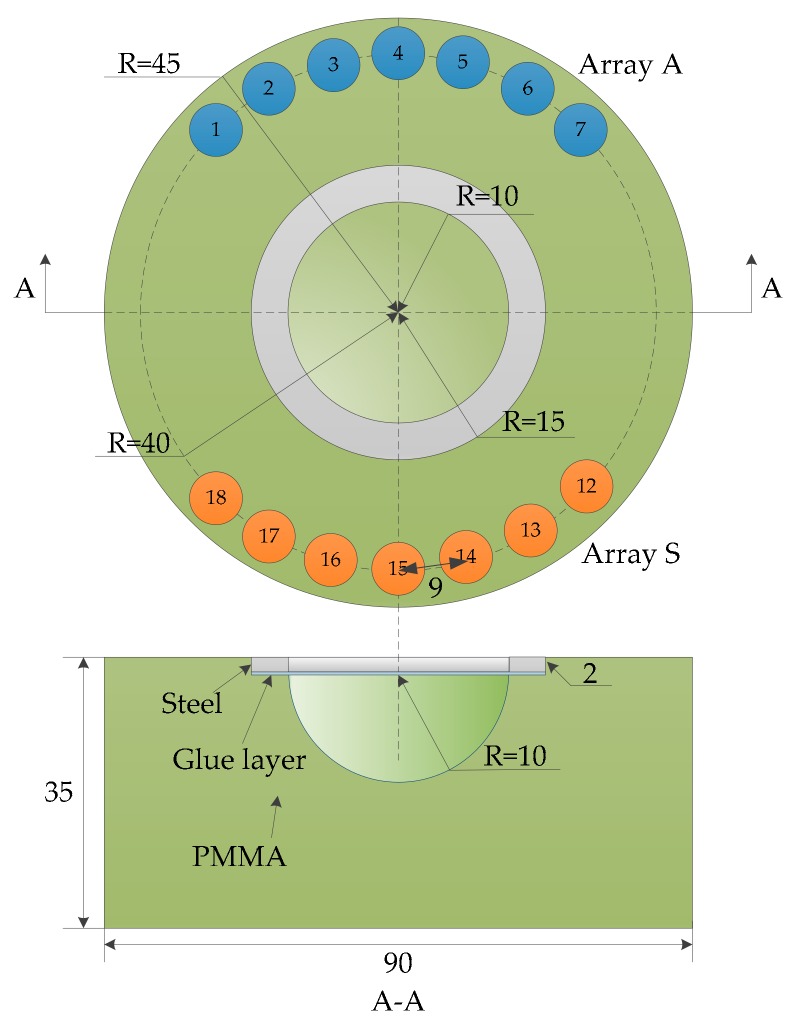
Dimensions of the circular bonded structures and sector sensor arrays.

**Figure 6 sensors-20-01265-f006:**
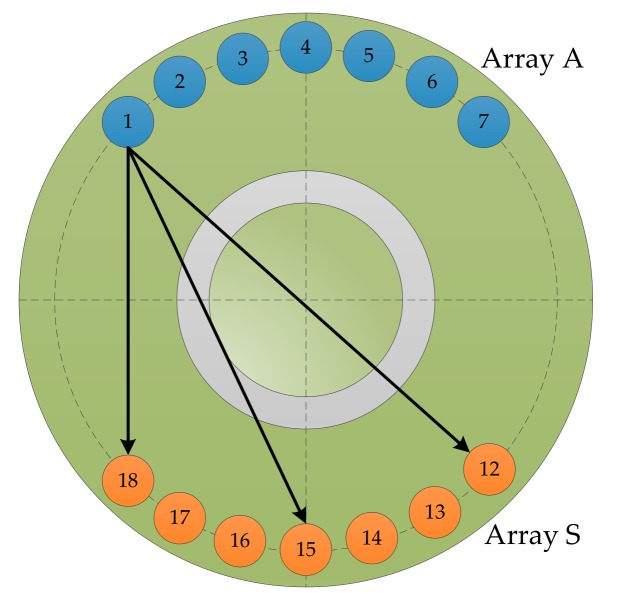
Three typical channels.

**Figure 7 sensors-20-01265-f007:**
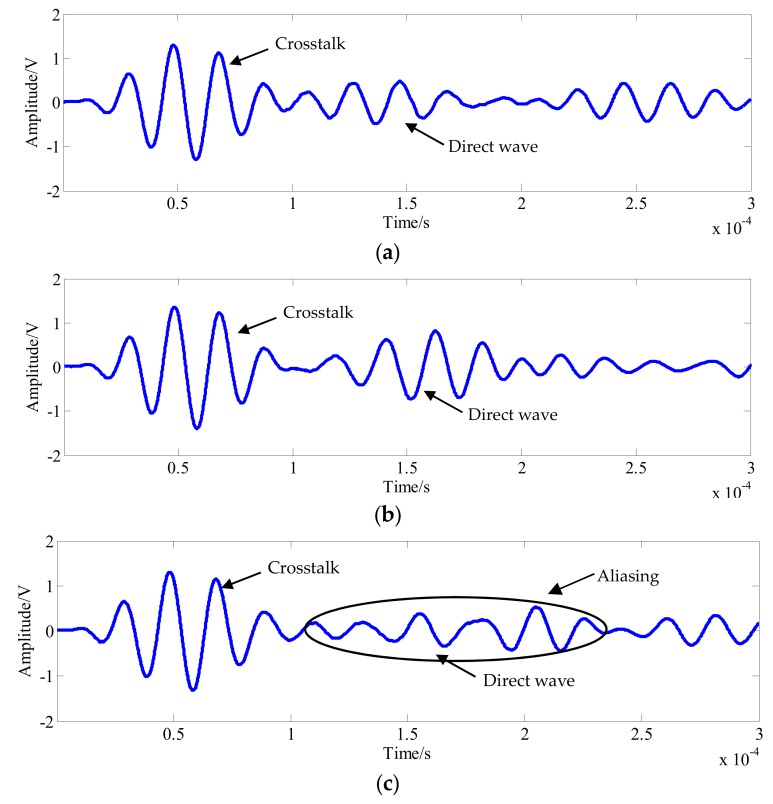
Signal waveforms of typical channels. (**a**) The signal of the PZT A_1_-PZT S_18_ channel. (**b**) The signal of the PZT A_1_-PZT S_15_ channel. (**c**) The signal of the PZT A_1_-PZT S_12_ channel.

**Figure 8 sensors-20-01265-f008:**
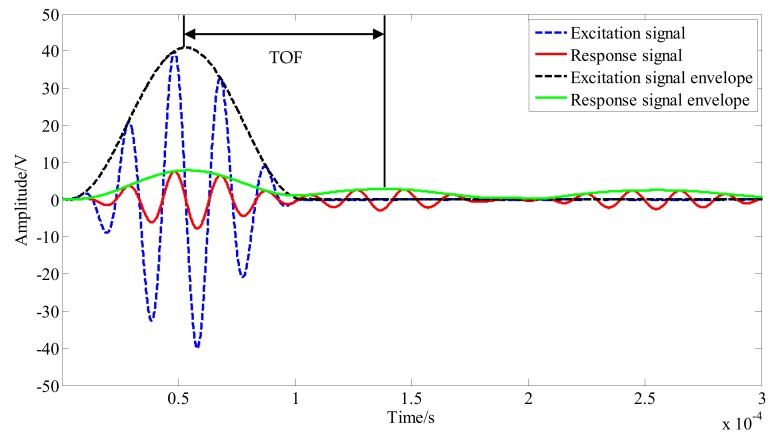
Calculation of flight time using the Shannon complex wavelet transform.

**Figure 9 sensors-20-01265-f009:**
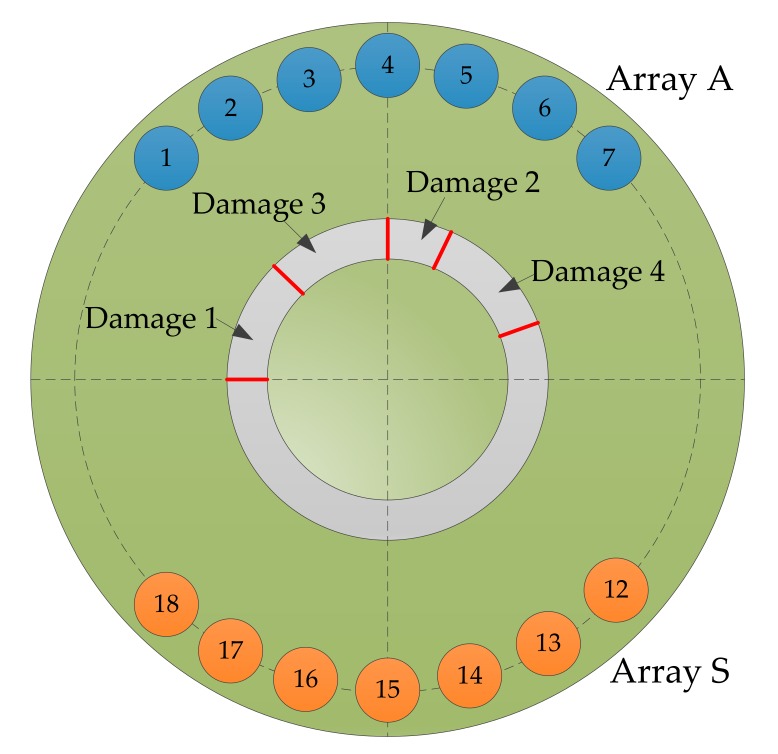
The location of damage to the structure.

**Figure 10 sensors-20-01265-f010:**
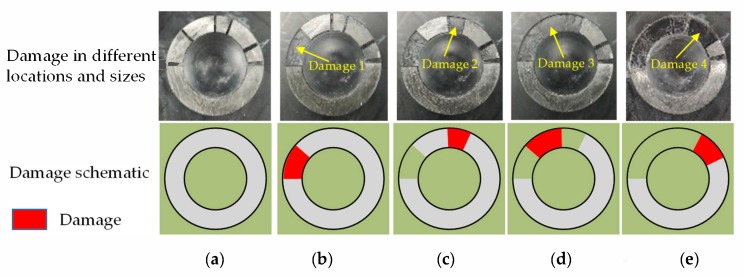
Details of damage. (**a**) Health. (**b**) Damage 1. (**c**) Damage 2. (**d**) Damage 3. (**e**) Damage 4.

**Figure 11 sensors-20-01265-f011:**
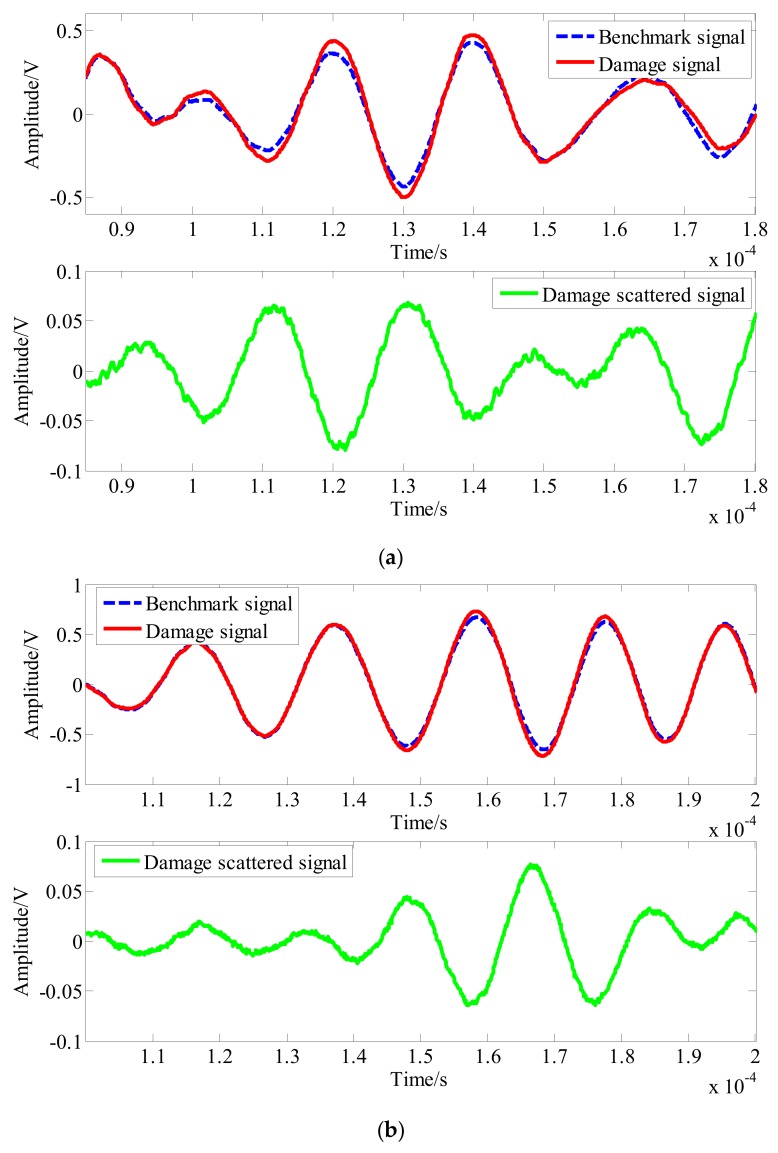
Effects of different sizes of damage on the signals. (**a**) The size of the damage is 10 × 5 mm^2^. (**b**) The size of the damage is 5 × 5 mm^2^.

**Figure 12 sensors-20-01265-f012:**
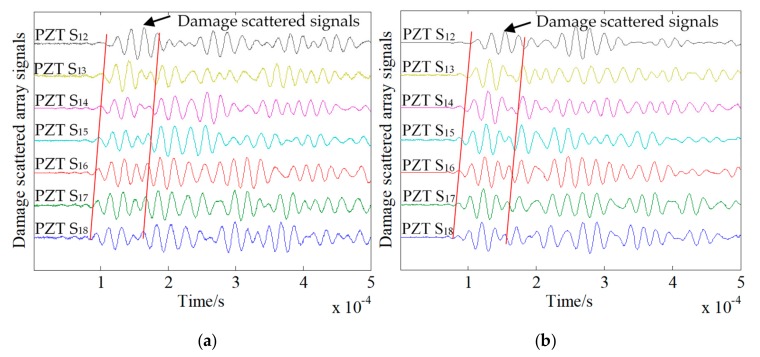
Damage scattering array signals before and after damage beamforming with a damage size of 10 × 5 mm^2^. (**a**) Before beamforming. (**b**) After beamforming.

**Figure 13 sensors-20-01265-f013:**
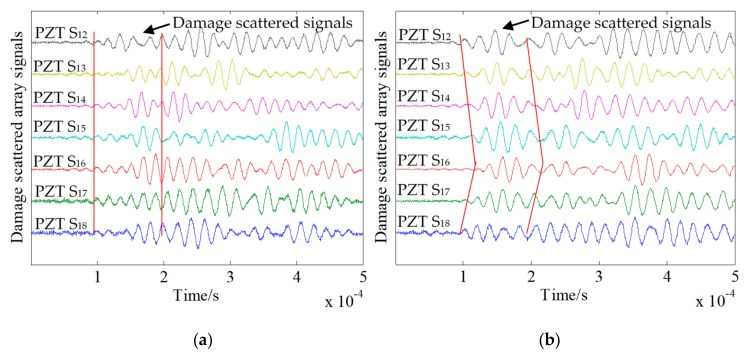
Damage scattering array signals before and after beamforming with a damage size of 5 × 5 mm^2^. (**a**) Before beamforming. (**b**) After beamforming.

**Figure 14 sensors-20-01265-f014:**
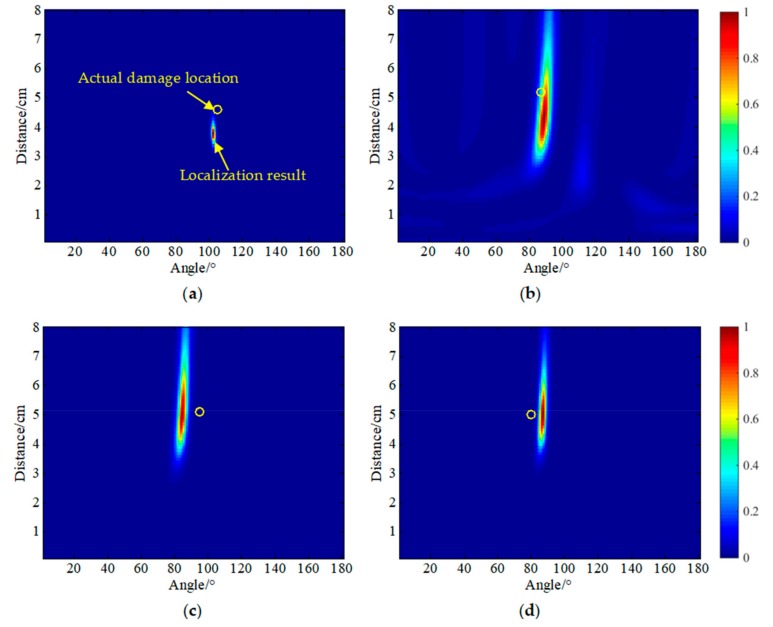
Damage localization results using the single transmitter MUSIC algorithm. (**a**) D1, (**b**) D2, (**c**) D3, (**d**) D4.

**Figure 15 sensors-20-01265-f015:**
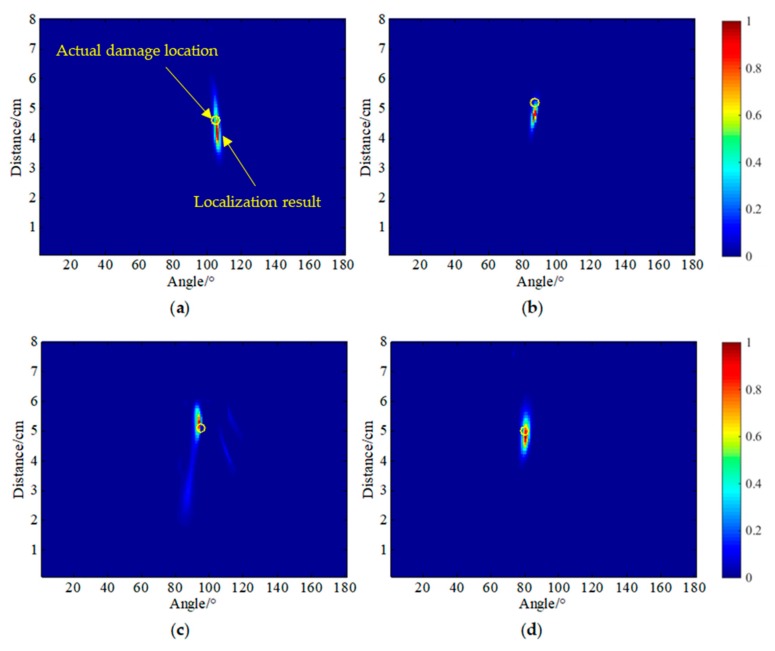
Damage localization results using the SPATB-MUSIC algorithm. (**a**) D1, (**b**) D2, (**c**) D3, (**d**) D4.

**Figure 16 sensors-20-01265-f016:**
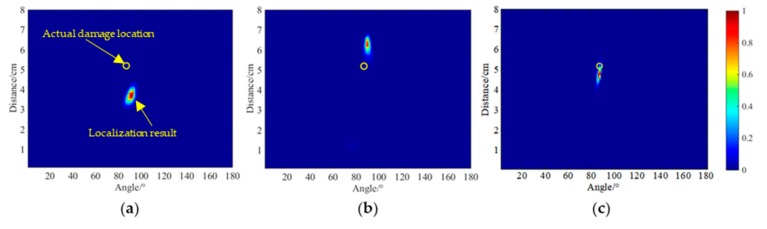
Results of damage localization for a damage size of 5 × 5 mm^2^. (**a**) Path imaging localization result. (**b**) Delay-sum imaging localization result. (**c**) STAPB-MUSIC localization result.

**Table 1 sensors-20-01265-t001:** Amplitudes and velocities of the three propagation paths.

Propagation Paths	Amplitude	Velocity
PZT A_1_-PZT S_18_	0.486 V	722 m/s
PZT A_1_-PZT S_15_	0.814 V	699 m/s
PZT A_1_-PZT S_12_	0.383 V	755 m/s

**Table 2 sensors-20-01265-t002:** Damage information.

Number	Size	Locations
D1	10 × 5 mm^2^	(105°, 4.6 cm)
D2	5 × 5 mm^2^	(87°, 5.2 cm)
D3	10 × 5 mm^2^	(95°, 5.1 cm)
D4	10 × 5 mm^2^	(80°, 5.0 cm)

**Table 3 sensors-20-01265-t003:** Damage localization results.

Number	Locations	Single Transmitter MUSIC			SPATB-MUSIC		
		Localization Results	Error		Localization Results	Error	
			Angle	Distance		Angle	Distance
D1	(105°, 4.6 cm)	(103°, 3.8 cm)	3°	0.8 cm	(105°, 4.3 cm)	0°	0.3 cm
D2	(87°, 5.2 cm)	(88°, 4.2 cm)	1°	1.0 cm	(86°, 4.7 cm)	1°	0.5 cm
D3	(95°, 5.1 cm)	(84°, 5.0 cm)	11°	0.1 cm	(94°, 5.3 cm)	1°	0.2 cm
D4	(80°, 5.0 cm)	(86°, 4.9 cm)	6°	0.1 cm	(80°, 4.8 cm)	0°	0.2 cm

**Table 4 sensors-20-01265-t004:** Localization results and errors for a damage size of 5 × 5 mm^2^.

Imaging Methods	Path Imaging	Delay-Sum Imaging	SPATB-MUSIC
*Error_d_*	1.5 cm	1.1 cm	0.5 cm
*Error_σ_*	18.8%	13.8%	6.3%

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
