# Peer review of "Sector Piezoelectric Sensor Array Transmitter Beamforming MUSIC Algorithm Based Structure Damage Imaging Method"

_sensors, 2020, doi:10.3390/s20051265_

Round 1

Reviewer 1 Report

The paper is interesting. It is good that the new method was compared with other ones to show its advantages. Some parts of the document needs clarifications:

What is p in equation (2)? The definition 'the time difference between the damage signal to the PZTi and the damage signal to the PZT0' is not clear. Do authors mean the difference of wave arrival? I think something is wrong in equation (3).  The segments: r, r are not parallel so Δri cannot be their difference. Please check equation (5).  '=' sign is there twice. What do authors mean by aliasing when describing fig. 7? For better illustration of the comparison I suggest to present the figures as (x,y) plots. in the current manuscript some of the results are plotted as (distance, angle) plots and some as (x,y) plots.

Reviewer 2 Report

Review report on the paper «Sector piezoelectric sensor array transmitter beamforming MUSIC algorithm based structure damage imaging method” by T. Fu et al.

The manuscript is well organized and includes a nice combination of theory and experimental finding that should fit well into the “Senor journals” scope of interests. The experimental investigation of a plexiglass/steel phantom points out benefits in terms of SNR and resolution, when combining MUSIC eigenvalue decomposition with classical beamforming. Due to these findings and currently a high interest of advanced damage detection methods in NDT, the paper is recommended for publication in the sensor journal after the following minor changes:

In the text on page 3. and also, in the caption for Fig. 1., it is said that the focusing method applies “in the near field”. However, the it is not clear why this approach requires a near field approximation, since phase steering normally works better in the far-field due to a larger overlap between the sound fields generated from the individual elements. The introduced near-field limitation should therefore be clarified in the manuscript.

From relative low velocities estimated for the phantom model (e.g. presented in Tab. 1), it looks like the exited wavefield is dominated by surface waves (Rayleigh waves), and not bulk waves that will propagate internally in the used materials. If, for example, longitudinal bulk waves are generated by the transducer, reflections of these high velocity waves might potentially interfere with the surface waves and reduce the accuracy of the proposed method. The manuscript is lacking information about different wave modes in the experiment, and strategies to e.g. reduce the contribution of bulk modes. I will therefore strongly recommend that such discussion is included.

If the considered problem is dominated by surface waves as the experimental data would indicate, it should be possible to consider it as a 2D problem. If this is the case, it will be somehow misleading to describe the wavefronts as spheres when they if fact, yield circles in a 2D model. I would therefore suggest that the authors reconsider the spherical model used in the middle of page 3, or at least clarify that the used model should be valid both for 2D and 3D waves.       

The meaning of the abbreviations “SPATB-MUSIC” and “SHM” probably have to be explained at their first occurrence in the main text (please check the journal rules here).

The words “…SNRS is…” in the abstract should be corrected to “…SNRs are…”

Reviewer 3 Report

The authors reported a structural damage imaging method based on the sector piezoelectric sensor array transmitter beamforming MUSIC algorithm, which adapts to small signal variations and improves the SNRs. The topic is of good interest to the sensor/actuator and SHM/NDE community. The paper is in general well-written. However, the following minor concerns should be addressed before it can be accepted for publication:

In the introduction, the authors should expand their discussion to provide the readers with more damage imaging approaches that address the interaction between the direct wave package and damage. For example, some recent progress by combining tomography with time-reversal technique, such as Wang, J.; Shen, Y. (2019) “An enhanced Lamb wave virtual time reversal technique for damage detection with transducer transfer function compensation”, Smart Materials and Structures, 28(8), 085017. The authors may include this in their introduction for the readers’ benefit. Will the direct wave packages aliased with the crosstalk signal and the boundary reflections influence the final imaging results? Please add a comment on this aspect. In Figure 13, why are the ToFs of the damage scattered signals different from each other before and after beamforming? The color scale of the imaging results should be provided.

Again, overall, the manuscript is well written. It should be considered for publication after one round of minor revision.
